# Metastatic SDH-Deficient GIST Diagnosed during Pregnancy: Approach to a Complex Case

Anas Chennouf [1], Elie Zeidan [1,*], Martin Borduas [2], Maxime Noël-Lamy [3], John Kremastiotis [4] and Annie Beaudoin [1,5]

1 Service of Gastroenterology, Centre Hospitalier Universitaire de Sherbrooke (CHUS), Sherbrooke, QC J1H 5H3, Canada
2 Department of Pathology, Centre Hospitalier Universitaire de Sherbrooke (CHUS), Sherbrooke, QC J1H 5H3, Canada
3 Department of Radiology, Centre Hospitalier Universitaire de Sherbrooke (CHUS), Sherbrooke, QC J1H 5H3, Canada
4 Department of Family Medicine, Université de Montréal, Montreal, QC H3T 1J4, Canada
5 Research Center, Centre Hospitalier Universitaire de Sherbrooke, Sherbrooke, QC J1H 5N4, Canada
* Correspondence: elie.zeidan@usherbrooke.ca

**Abstract:** Gastrointestinal stromal tumors (GISTs) account for 1% of GI neoplasms in adults, and epidemiological data suggest an even lower occurrence in pregnant women. The majority of GISTs are caused by KIT and PDGFRA mutations. This is not the case in women of childbearing age. Some GISTs do not have a KIT/PDGFRA mutation and are classified as wild-type (WT) GISTs. WT-GIST includes many molecular subtypes including SDH deficiencies. In this paper, we present the first case report of a metastatic SDH-deficient GIST in a 23-year-old pregnant patient and the challenges encountered given her concurrent pregnancy. Our patient underwent a surgical tumor resection of her gastric GIST as well as a lymphadenectomy a week after induction of labor at 37 + 1 weeks. She received imatinib, sunitinib as well as regorafenib afterward. These drugs were discontinued because of disease progression despite treatment or after side effects were reported. Hence, she is currently under treatment with ripretinib. Her last FDG-PET showed a stable disease. This case highlights the complexity of GI malignancy care during pregnancy, and the presentation and management particularities of metastatic WT-GISTs. This case also emphasizes the need for a multidisciplinary approach and better clinical guidelines for offering optimal management to women in this specific context.

**Keywords:** GIST; gastrointestinal stromal tumor; malignancy; pregnancy; SDH deficient; abdominal mass

## 1. Introduction

Gastrointestinal stromal tumors (GISTs) are mesenchymal neoplasms originating from the interstitial cells of Cajal. GISTs are the most common soft tissue sarcoma of the GI tract but remain fairly uncommon, with an incidence of 10–15 cases per million per year. The median age of diagnosis is 64 years old [1]. Considering the age distribution of the disease, a diagnosis of GIST during pregnancy is exceedingly rare, making its management even more complex for clinicians. GISTs during young adulthood are more commonly associated with SDH mutations, making them less responsive to tyrosine kinase inhibitors (TKIs), the mainstay of treatment in advanced GISTs.

To date, there have been less than twenty reported cases of GISTs during pregnancy and there is currently no guideline for management in this specific context. We hereby present the first case of an SDH-deficient GIST in a pregnant patient and present our approach to this complex case.

## 2. Case Presentation

A 23-year-old female was referred to our GI clinic at 23 weeks of pregnancy for a palpable epigastric mass. She first noticed this mass a year ago after her first pregnancy. She was otherwise asymptomatic. Her personal as well as familial past medical histories were unremarkable, and her pregnancy was otherwise eventless. Her physical examination was normal except for the palpable epigastric mass. Laboratory work-up showed a hemoglobin level of 110 g/L (N 120–160) with an MCV of 66.5 (N 80–100). The patient had already been on optimal iron supplementation for 3 months. She was not taking any other medication.

An abdominal unenhanced MRI showed a large well-defined submucosal polylobate antral gastric mass of the size $7.6 \times 4 \times 5$ cm. The mass was solid, with cystic components. Lymph node involvement was suspected. These lymph nodes measured up to 1 cm on the short axis. These findings were consistent with a GIST.

An EUS confirmed that the pre-pyloric solid mass originated from the muscularis mucosa. The mass had two mucosal ulcers on its surface. The endoscopic appearance was suggestive of a GIST, but a neuroendocrine tumor could not be excluded. Chromographin A was mildly elevated at 118 ug/L, and urinary 5-HIAA was normal. The patient had no symptoms suggestive of carcinoid syndrome.

A multidisciplinary team (MDT) including surgeons, gastroenterologists, oncologists, radiologists, obstetrician-gynecologists (OBGYNs) and maternal–fetal medical (MFM) specialists discussed the case to offer the best medical options to the patient. After assessing the maternal/fetal surgical risk and the risk of tumor malignity (size, location and evolution), a decision was made to proceed to surgery after an induction of labor between 34 and 37 weeks to maximize fetal viability and to avoid bleeding complications. Performing a tumor biopsy was not considered given the potential risk of dissemination and hemorrhage. An abdominal ultrasound planned 3 weeks later showed no tumor growth.

The patient was induced at 37 + 1 week. The baby was born with no immediate complications with an Apgar score of 9-9-10. She was discharged 2 days postpartum and was readmitted again a week later for her elective surgery. A preoperative abdominal CT with contrast was performed (Figure 1). The mass seemed to have progressed, now measuring 8.3 cm. A 10 mm enhancing retroperitoneal lymph node anterior to the aortic bifurcation suspicious for metastasis was noted. A 10 mm enhancing retroperitoneal lymph node anterior to the aortic bifurcation was visualized, with a suspicion of metastasis. Partial gastrectomy and a Billroth type I reconstruction were performed. During surgery, macroscopic lymph nodes were detected across the gastro-epiploic artery. Consequently, a lymphadenectomy extending to the hepatic pedicle was completed.

Pathology was consistent with a multifocal, invasive, gastric and duodenal GIST with a predominant spindle cell component (Figures 2 and 3). The size sample could not be measured with accuracy, but it was more than 5 cm. The mitotic index was <5 mitosis/5 mm$^2$. Immunohistochemical studies were highly positive and diffuse for CD117. Surgical margins were positive, and there was a lymphovascular invasion. Metastasis was found in 5 of the 11 excised lymph nodes, including 2 lymph nodes localized on the greater omentum. The tumor was classified as a pT3(m)N1 according to the AJCC Cancer Staging Manual 8th edition. Molecular studies were negative for KIT and PDGFRA mutations.

Three weeks later, a post-operative FDG-PET scan was performed and showed a hypermetabolic 9 mm lymph node at the aortic bifurcation with an SUV of 5.9 and a discretely hypermetabolic mass adjacent to the distal duodenum, with an SUV of 2.4 that was equivocal for metastasis. There were no other suspicious findings.

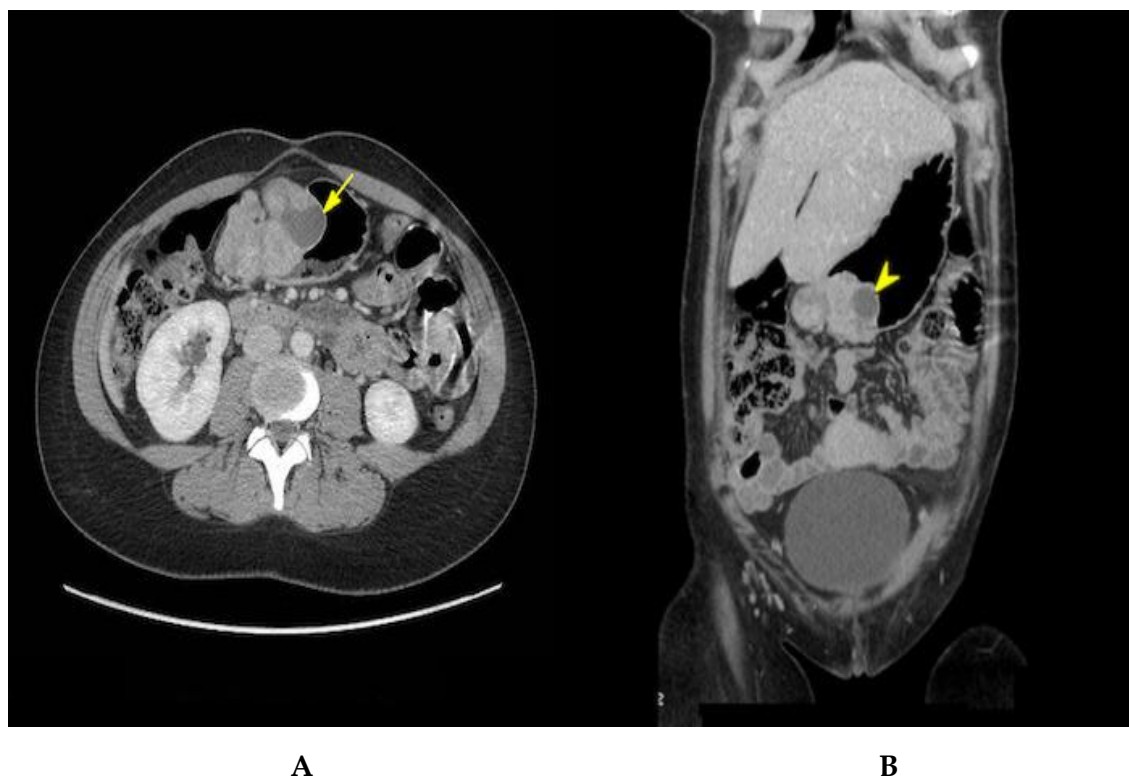

**A** **B**

**Figure 1.** Axial (**A**) and coronal (**B**) contrast-enhanced CT view showing an 8.3 cm polylobate submucosal lesion with cystic components (yellow arrows) consistent with a GIST.

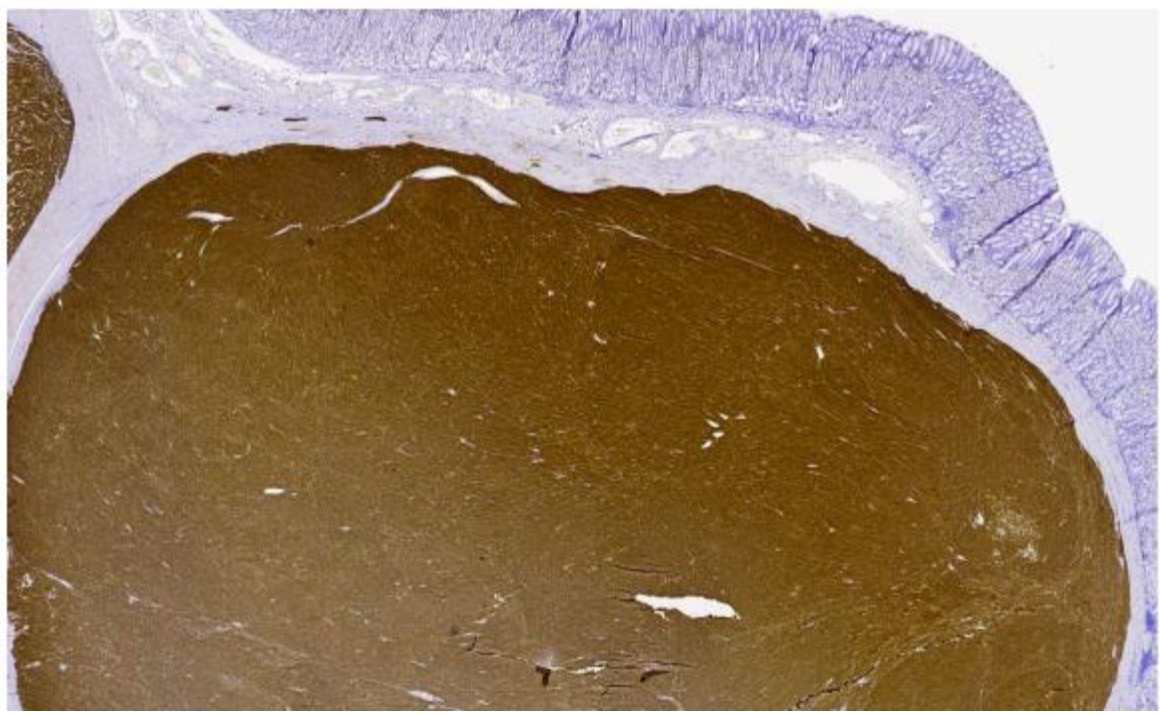

**Figure 2.** Pathological cut of gastric GIST with immunohistochemistry stain for CD117/c-kit.

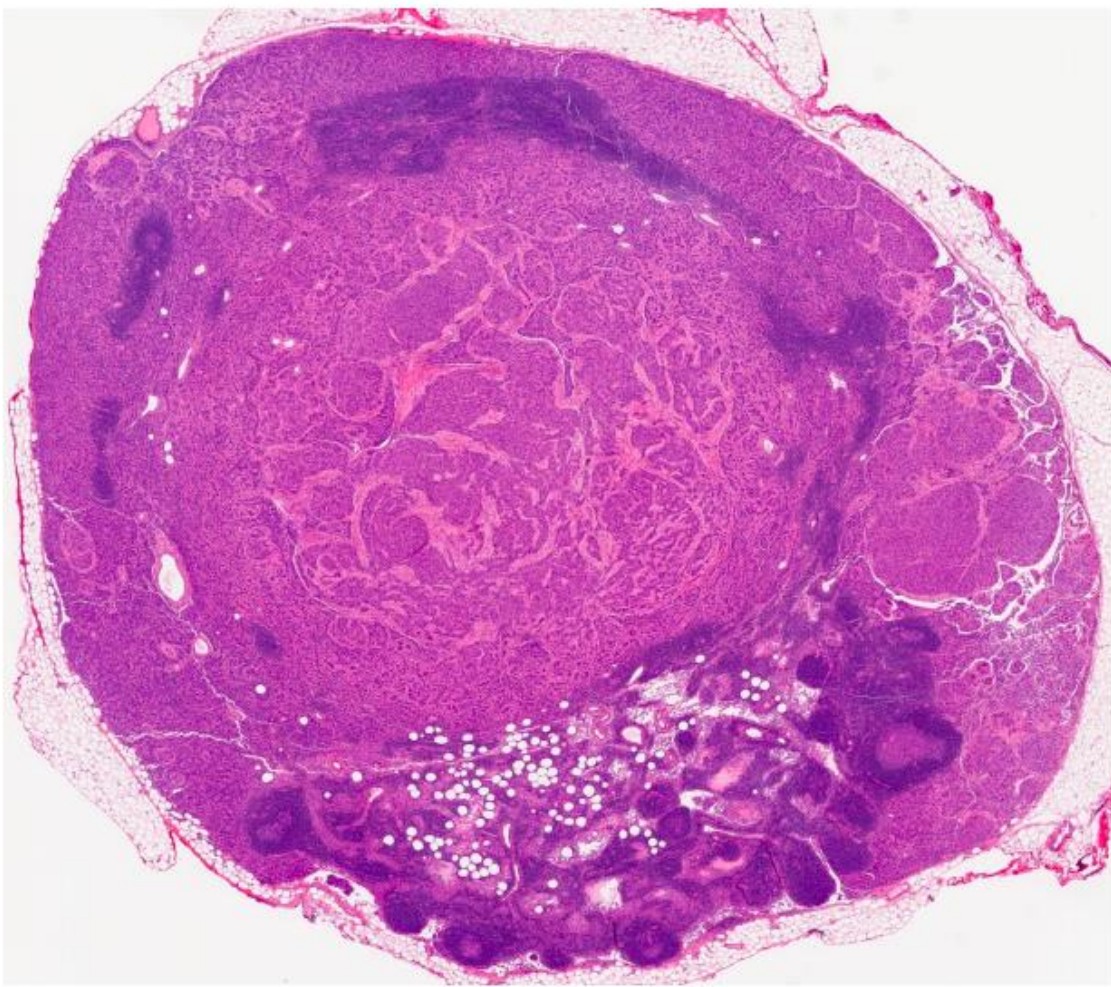

**Figure 3.** HE stains showing GIST lymph node metastasis.

A diagnosis of metastatic GIST was confirmed, and treatment was initiated with imatinib at a standard dose of 400 mg. The patient was also started on contraceptives. A genetic and molecular referral was requested to assess for wild-type (WT) GISTs. The genetic panel revealed a heterozygote mutation in the SDHC gene. Even though a WT-GIST was confirmed, imatinib was continued. In fact, her 4-month follow-up FDG-PET scan showed a stable disease while receiving imatinib.

Considering the known association between SDH deficiencies and pheochromocytomas and paragangliomas (PPGLs), a neck and thorax MRI and a 68Ga DOTATE PET scan were performed and showed no evidence of PPGLs. Urinary and plasmatic catecholamines were normal.

A 9-month follow-up FDG-TEP scan (Figure 4) after the introduction of imatinib showed disease progression with new liver metastasis. Therefore, sunitinib was started at the standard dose of 50 mg. Despite a 3-month course of this second-line treatment, further disease progression led us to initiate regorafenib at 160 mg. Unfortunately, the patient developed a severe morbilliform pancorporal rash 10 days after the introduction of regorafenib. She was treated with oral prednisone and her rash disappeared a few days later. Upon follow-up, once her skin rash resolved, a trial of 40 mg of regorafenib was tried. However, she developed a similar rash 4 h after taking the medication. Lastly, regorafenib was discontinued and ripretinib was started at 150 mg once a day. Her primary disease progressed at this usual dose, leading to a dose increase to 150 mg twice daily. The metastatic burden of her disease has remained stable under this fourth-line treatment for nearly a year at the current regimen. Her ongoing treatment is well tolerated. She has

developed moderate palmoplantar hyperkeratosis induced by ripretinib, which is currently managed with topical treatment. Figure 5 summarizes the patient's treatments.

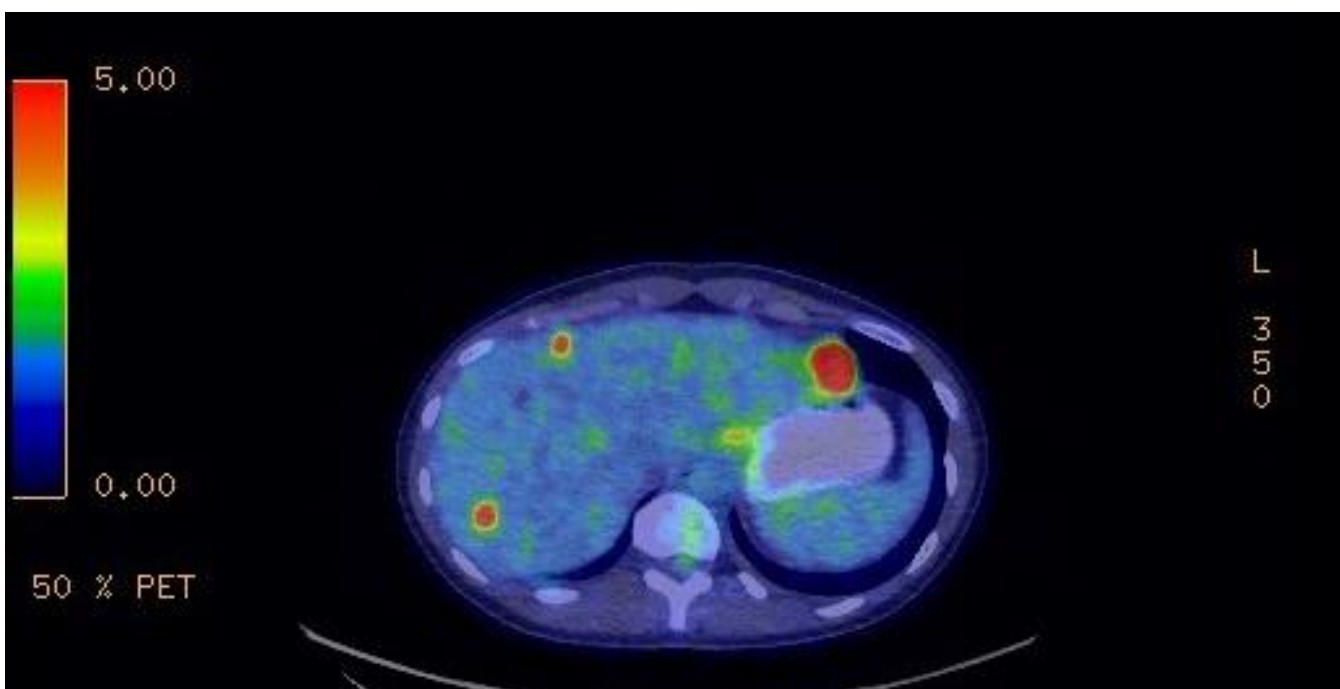

**Figure 4.** Nine-month follow-up FDG-TEP scan after introduction of imatinib, showing disease progression with new liver metastasis.

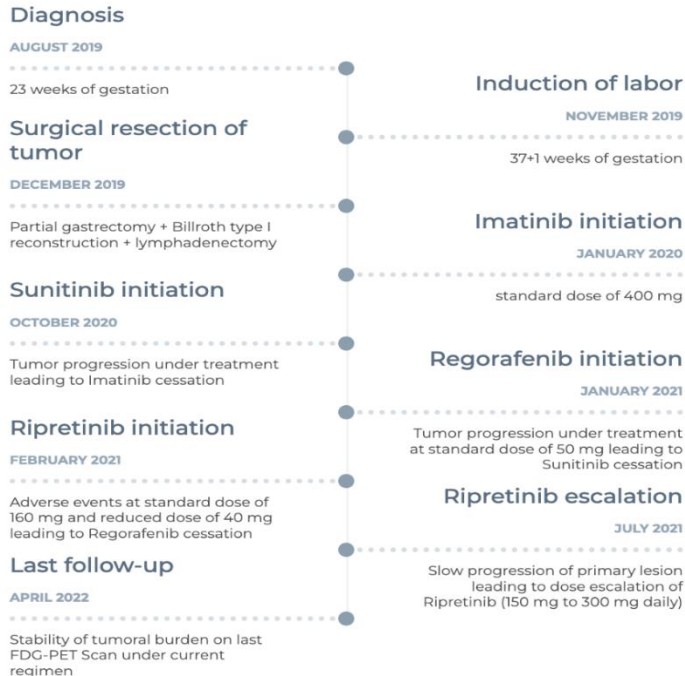

**Figure 5.** Timeline of the patient's diagnosis and treatments.

## 3. Discussion

A molecular diagnosis of GIST can be made based on immunohistochemistry studies. In fact, the expression of CD117 and DOG1 allows for the diagnosis of approximately 99% of

GISTs [2]. Furthermore, mutations in the KIT or PDGFRA genes are present in most GISTs (85%) diagnosed in adults. These mutations cause the activation of signaling pathways for tumor proliferation [3]. The discovery of these proto-oncogenes' role has transformed GIST management, which was widely known to be radio- and chemo-resistant. Imatinib, a selective inhibitor of the KIT protein tyrosine kinase (TKI), has become a standard regimen for the management of advanced and metastatic GISTs, inducing an objective response in more than 50% of patients [4] and dramatically improving survival in treated patients [5].

Some GISTs do not have a KIT/PDGFRA mutation and are consequently classified as WT-GISTs. WT-GIST includes many molecular subtypes including SDH deficiencies, SDHx mutations, NF1 mutations and BRAF mutations. They can appear sporadically or as part of a syndrome. A large study by the National Institutes of Health (NIH) GIST clinic including 95 patients with WT-GISTs showed that the majority of these tumors (88%) had an SDH-related anomaly (deficiency, mutation or SDHC promoter methylation) [6]. Clinically, these tumors are different from KIT/PDGFRA-mutant GISTs. In fact, while GISTs rarely affect young patients, WT-GISTs more often affect children and young adults (median age <30 years old) as well as females at a ratio of 2:1 [7]. They often present as multinodular and multifocal tumors, appearing almost exclusively in the stomach with a tropism for the antrum and distal stomach. Histologically, they are characterized by epithelioid or mixed cells with frequent lymphovascular invasion and lymph node metastasis [8]. Studies seem to suggest that SDH-deficient GISTs have an indolent clinical course with a median survival of >10 years despite high local recurrence rates and metastases. A study by Miettinen et al. showed that 65% of patients with metastatic SDH-deficient GISTs survived 2–43 years after surgery and 15% died of the disease, with a median survival of 9 years [9]. Considering their rare occurrence, prognostic and risk stratification of SDH-deficient GISTs remains to be elucidated. Current NIH health risk stratification tools do not seem to correlate with disease progression. In fact, in a study by Mason and Hornick, 63% of patients categorized as very low or low risk eventually developed distant metastasis during follow-up [10]. This emphasizes the need for selecting better prognostic factors.

Surgical excision remains the mainstay of treatment for localized WT-GISTs. Lymphadenectomy is often required because of tumor invasion. Quality of surgical resection (presence of negative margins) and surgery type (partial vs. total excision) do not seem to impact the event-free survival (EFS) in these patients [11]. More interestingly, repeated resection after the initial surgery is significantly associated with a decrease in post-operative EFS and should be limited to symptomatic (bleeding or obstruction) recurrent disease [11]. Management can be different in cases of locally advanced GISTs. A prospective study evaluating the neoadjuvant use of imatinib in locally advanced GISTs was conducted on 51 patients. This study aimed to evaluate the optimal duration of neoadjuvant imatinib before surgery. It concluded that the median time of maximum shrinkage was 6.1 months [12]. Furthermore, a retrospective study in 2018 concluded that progression-free survival (PFS) and overall survival (OS) were prolonged when metastatic GISTs with responsive disease or stable disease were treated with imatinib followed by metastasectomy compared to imatinib alone [13].

Although imatinib is very effective at treating KIT/PDGFRA-mutant GISTs, SDH-deficient GISTs are largely resistant to TKIs due to the absence of gain-of-function tyrosine kinase mutations [14]. The response rate to imatinib is believed to be around 2% [15]. However, data seem to show a superiority of sunitinib for imatinib-resistant tumors with a response rate of around 10% [16], an effect possibly attributable to its anti-VEGF target and anti-angiogenesis characteristics. This could partially explain its efficacity against SDH-deficient GIST. Despite these results, current NCCN guidelines still suggest a therapeutic trial of imatinib for KIT-negative GISTs while emphasizing the importance of careful and close follow-up. Regorafenib is currently the third-line treatment for WT-GIST. However, a phase II–III study on regorafenib involving WT-GISTs (including only six SDH-deficient GIST) showed that only two patients had a partial response to this drug. The rest of these patients had stable diseases [15]. Lastly, ripretinib is currently the fourth-line treatment for

GISTs. It showed inhibitions of progression of WT-GIST in preclinical testing [15]. Our case report also highlights the importance of considering dose escalation in the case of disease progression. In fact, our patient responded to a fourth-line treatment (ripretinib) at double the standard dose. To our knowledge, there are no specific studies on WT-GIST treated with ripretinib. The systemic treatment approach remains a controversial issue in WT-GIST management, and clinical trials studying the effect of experimental targeted therapies and alkylating agents are currently in progress [17,18].

Although the majority of GISTs appear to be sporadic, SDH-deficient GISTs can be syndromic in around 5% of cases [16]. Carney–Stratakis syndrome is characterized by the development of GIST and paragangliomas. It is attributed to germline mutations in one of the SDH gene subunits (SDHA, SDHB, SDHC or SDHD). This syndrome is generally hereditary. Another syndrome associated with SDH-deficient GIST is the Carney triad. It consists of a combination of GIST, paraganglioma and pulmonary chondroma. It is believed to be caused by epigenetic hypermethylation of SDHC promoter rather than an SDH mutation. This makes it less susceptible to being transmitted hereditarily [8].

Patients with germline mutations of an SDHx gene are also susceptible to developing pheochromocytomas and paragangliomas (PPGLs). It is responsible for 30–40% of hereditary PPGL cases [19]. Studies seem to suggest a heterogeneous pattern of presentation and penetrance depending on the SDH subunit affected. For example, mutations in the SDHB gene are associated with a higher risk of mortality and neoplastic potential [20]. Consequently, biochemical and imaging surveillance is recommended in this population. However, there is currently little data on the natural evolution of SDHC-related tumors (as seen in our patient), and recommendations for long-term surveillance are rather scarce [14].

Malignancy during gestation involves major management implications that should be anticipated by clinicians. First, the optimal timing of delivery should be discussed with a multidisciplinary team including surgeons, oncologists and MFMs. Delivery should be delayed to the 35th–37th weeks, when possible, to avoid premature and neonatal complications. In a systematic review of 12 cases of GISTs occurring during pregnancy, the median gestational week at delivery was 36 weeks. A total of 11/12 patients were delivered by C-section, including one case due to fetal distress [21]. All the newborns were healthy at follow-up.

Moreover, using imatinib in advanced disease should be carefully assessed given experimental data showing teratogenic effects [22]. Pye et al. showed that 9.6% of fetuses exposed to imatinib had birth defects, including skull and spine malformations, exomphalos, hydrocephalus, heart defects and cerebellar hypoplasia [23]. This study included 180 women. Most of them were exposed to imatinib during their first trimester, and there were no cases of birth defects detected when imatinib was strictly taken during the second and third trimesters. Finally, considering the hereditary pattern of SDH-deficiency-related neoplasms, patients with WT-GISTs should be referred for genetic evaluation and counselling as part of the patient's and infant's medical management.

## 4. Conclusions

To our knowledge, this is the first case report describing the management of a metastatic and refractory SDH-deficient GIST diagnosed during pregnancy. Considering the preponderant incidence of WT-GISTs in young women of childbearing age, a diagnosis of SDH-deficient GISTs should be considered in pregnant women presenting with a gastric mass, particularly when investigations support a diagnosis of GIST and do not identify KIT or PDGFRA mutations. This case report also highlights that, in the case of progression, doubling the dose of ripretinib is an interesting option to try. Finally, this paper emphasizes the importance of a multidisciplinary approach and the need for better clinical data and guidelines for offering optimal management of malignancy in this specific context.

**Author Contributions:** Conceptualization, A.B., E.Z. and A.C.; validation, A.B.; resources, A.C. and E.Z.; writing—original draft preparation, A.C., E.Z., A.B., M.N.-L. and M.B.; writing—review and editing, J.K., E.Z. and A.C.; visualization, J.K., A.C. and E.Z.; supervision, A.B.; project administration, A.B. All authors have read and agreed to the published version of the manuscript.

**Funding:** This research received no external funding.

**Institutional Review Board Statement:** The study was approved by CIUSSS Estrie-chus' Research ethics Committee (protocol code 2023-4773).

**Informed Consent Statement:** Informed consent was obtained from all subjects involved in the study.

**Data Availability Statement:** The data presented in this study are available on request from the corresponding author.

**Conflicts of Interest:** The authors declare no conflict of interest.

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
