# Peer review of "Metastatic SDH-Deficient GIST Diagnosed during Pregnancy: Approach to a Complex Case"

_curroncol, doi:10.3390/curroncol29080468_

Round 1
Reviewer 1 Report
The authors reported a case of SDH-deficient GIST during pregnancy. This case is suitable for publication but some issues should be addressed and revised.
Major commets:
1. My major concern is that this manuscript is not well-written and full of grammar mistakes. In addition, this seems like a medical record rather than academic writing.
2. Please showed the PET scan for metastatic GISTs.
3. Please check the decimal points in whole manuscript and make them consistent.
4. A neck and thorax MRI and a 68Ga DOTATE PET Scan showed no evidence of paraganglioma. Urinary and plasmatic catecholamines were normal. Why ?
5. She progressed at this usual dose. However, the metastatic burden of her disease has remained stable under this fourth line treatment for almost a year now while receiving 104 ripretinib 150mg twice a day. Why did she experience progression but also stable disease ?
6. I suggested added some comments and discussion of some points to make this article more comprehensive.
May consider cite the referece in terms of molecular diagnosis of GIST. (PMID: 31100836)
May discuss the role of local surgery in addition to RTO (PMID: 33976743, PMID: 28448384)
May discus the role of RTO in neoadjuvant setting in addition to TKIs (PMID: 30934606)
7. Why did authors keep the figures in the Appendix A ? No figures in the main manuscript is strange. The legends of figures are too rough. The authors should describe more for the figure. The tumor size looks like 5-6 cm only in Figure A1.
8. Please make a figure to summarize the clinical course including surgery and treatment.
9. Why didn’t you do Bx to confirm the diagnosis and evaluate the risk in the beginning of multidisciplinary team discussion. Biopsy and genetic test may provide more information for the risk evaluation. The authors should discussion.
10. The authors did not provide IHC staining for SDHB to confirm SDH-deficient GIST.
Minor comments
1. What dose OBGYNs stand for ? all the abbreviations should be presented with full name for first appearance. Please check it.
2. 2 lymph nodes localized on the greater omentum à metastatic GIST ?
3. She progressed ? Why can she progress ? (Line 97)
4. What is target-organ damage ?
5. She was rechallenged ? (Line 100)
6. Develop (Line 101) should be past tense
7. switched for ? (Line 102)
8. Line 130 “only 35%” should depend on follow-up time. The longer f/u should be higher mortality rate.
Reviewer 2 Report
The authors present a case report of the SDH-deficient GISTs' in a pregnant woman. Moreover, the presented report describes the nature and the management of the disease, which, in such a patient, is extremely rare. The authors emphasize the necessity of a multidisciplinary approach and development of new guidelines for treatment in this specific context.
To begin with, the authors touch on an issue which has not been reported at all in literature. Therefore the topic is novel and the discussion highlights a significant issue in current guidelines, which may require their re-evaluation.
The manuscript is divided into the introduction, case presentation and discussion followed by a conclusion (mistake in section numbers 1,2,4 and 5; no section no.3). This division allows the reader to follow the authors easily.
My only concern is the fact that the tumor was first palpated a year before and the title "Metastatic SDH-deficient GIST occurring during pregnancy: approach to a complex case" indicates correlation, while it could only be coincidence. I would consider rephrasing the title. My suggestion is: "Management of metastatic SDH-deficient GIST diagnosed during pregnancy: approach to a complex case".
Overall, this case report is well presented, interesting and original. The references are fine.
Round 2
Reviewer 1 Report
Authors have revised this manuscript and current form is suitable for publication.